# Role of Mitochondrial ROS for Calcium Alternans in Atrial Myocytes

**DOI:** 10.3390/biom14020144

**Published:** 2024-01-24

**Authors:** Yuriana Oropeza-Almazán, Lothar A. Blatter

**Affiliations:** Department of Physiology and Biophysics, Rush University Medical Center, 1750 W. Harrison St., Chicago, IL 60612, USA; maria_y_oropezaalmazan@rush.edu

**Keywords:** atrial fibrillation, Ca alternans, cellular redox regulation, electron transport chain, excitation–contraction coupling, mitochondrial dysfunction, mitochondrial ROS

## Abstract

Atrial calcium transient (CaT) alternans is defined as beat-to-beat alternations in CaT amplitude and is causally linked to atrial fibrillation (AF). Mitochondria play a significant role in cardiac excitation–contraction coupling and Ca signaling through redox environment regulation. In isolated rabbit atrial myocytes, ROS production is enhanced during CaT alternans, measured by fluorescence microscopy. Exogenous ROS (tert-butyl hydroperoxide) enhanced CaT alternans, whereas ROS scavengers (dithiothreitol, MnTBAP, quercetin, tempol) alleviated CaT alternans. While the inhibition of cellular NADPH oxidases had no effect on CaT alternans, interference with mitochondrial ROS (ROS_m_) production had profound effects: (1) the superoxide dismutase mimetic MitoTempo diminished CaT alternans and shifted the pacing threshold to higher frequencies; (2) the inhibition of cyt *c* peroxidase by SS-31, and inhibitors of ROS_m_ production by complexes of the electron transport chain S1QEL1.1 and S3QEL2, decreased the severity of CaT alternans; however (3) the impairment of mitochondrial antioxidant defense by the inhibition of nicotinamide nucleotide transhydrogenase with NBD-Cl and thioredoxin reductase-2 with auranofin enhanced CaT alternans. Our results suggest that intact mitochondrial antioxidant defense provides crucial protection against pro-arrhythmic CaT alternans. Thus, modulating the mitochondrial redox state represents a potential therapeutic approach for alternans-associated arrhythmias, including AF.

## 1. Introduction

Atrial fibrillation (AF) is the most common cardiac arrhythmia and is a leading cause of heart failure (HF) and stroke and increases the risk of death. AF currently affects 3 to 6 million people in the United States and projections for 2050 indicate a prevalence of 6 to 16 million [1]. Cardiac alternans precedes and provides an arrhythmogenic substrate for AF and refers to beat-to-beat alternations in action potential duration, Ca transient amplitude and contraction strength [2,3]. A general paradigm predicts that impaired cytosolic Ca sequestration facilitates alternans, whereas factors increasing Ca sequestration protect against alternans [4]. In atrial myocytes, lacking a transverse (t) tubule system, Ca-induced Ca release (CICR) from junctional sarcoplasmic reticulum (SR) Ca stores is initiated by action potential-dependent Ca entry through voltage-gated Ca channels in the cell periphery from where CICR from non-junctional SR propagates in a Ca wave-like fashion centripetally to the cell center [5,6,7]. Ca uptake by mitochondria packed between the myofilaments can modulate the CaT amplitude and blunt the velocity of the centripetal Ca propagation [6], thus underscoring the role of mitochondria for atrial Ca signaling during excitation–contraction coupling (ECC). Moreover, mitochondrial Ca uptake and sequestration couple ECC energy demands with oxidative metabolism for ATP production. Oxidative metabolism is linked to the electron flux along the electron transport chain (ETC), comprised of respiratory Complexes I–IV, which catalyze the electron transport from NADH and FADH_2_ to O_2_ for the proton translocation across the mitochondrial inner membrane to generate the electrochemical gradient (ΔΨ_m_) which fuels ATP synthesis [8,9].

Mitochondria are an important source of reactive oxygen species (ROS_m_) in the form of superoxide (O_2_^·−^) and hydrogen peroxide (H_2_O_2_) from sites associated with substrate catabolism and the ETC, where Complexes I and III are recognized as the primary sites of ROS_m_ production [10,11]. Additionally, ROS are also produced within mitochondria by other proteins such as NADPH oxidase 4 (NOX4), monoamine oxidases isoforms A and B, p66^Shc^ and uncoupled mitochondrial nitric oxide synthase [12,13,14]. O_2_^·−^ is rapidly dismuted to H_2_O_2_ by the mitochondrial superoxide dismutase (SOD), and H_2_O_2_ is further eliminated by glutathione peroxidase (GPX) and peroxiredoxin (PRX)/thioredoxin (TRX-2) systems, which are regenerated by NADPH [15,16]. NADPH is regenerated by the Krebs cycle intermediates and the nicotinamide nucleotide transhydrogenase (THD) which uses NADH as a substrate [17,18].

ROS are also produced by cytosolic enzymes such as NADPH oxidases, lipoxygenase and cyclooxygenase, cytochrome P450s, xanthine oxidase and nitric oxide synthases [19]. While ROS act as signaling molecules in a variety of physiological processes, an imbalance in ROS generation and clearance can result in oxidative stress [20]. Oxidative stress may lead to the pathological activation of signaling pathways and oxidation of lipids and ECC proteins, modifying their activity and potentially impairing intracellular Ca handling. Such disturbances of cellular Ca signaling have been linked to the pathogenesis of arrhythmias [21,22]. While multiple studies have demonstrated the effects of ROS and oxidative stress on ventricular arrhythmias and sudden cardiac death [23,24,25,26,27,28], the relationship between the mitochondrial redox state and AF has been studied less. An investigation by Yoo et al. (2018) suggested that the oxidation of Ca/calmodulin-dependent kinase II by ROS_m_ promotes AF in a canine model of HF [29]. Xie et al. (2015) showed that mutant mice with leak-prone ryanodine receptor type-2 (RyR2) Ca-release channels produce higher levels of ROS_m_, which, in turn, promote RyR2 oxidation and increase AF susceptibility. The role of ROS_m_ production for the development of AF was addressed by overexpressing human catalase targeted to mitochondria which decreased AF prevalence [30]. Nonetheless, mechanistic data on the role of ROS_m_ and AF are still scarce. Moreover, the relative contributions of different sites of ROS generation, and specifically ROS_m_, to the development of CaT alternans has not yet been explored. For the regulation of the cellular redox environment, mitochondria not only contribute as a ROS source, mitochondrial antioxidant defense mechanisms also regulate cytosolic ROS handling, giving mitochondria a prominent role as a cellular ROS stabilizer [18,31,32].

The goal of this study was to determine whether ROS_m_ production determines the susceptibility of rabbit atrial myocytes to develop pacing-inducing CaT alternans. We first explored the relationship between ROS production and CaT alternans development. Next, we investigated the effects of the modulation of the intracellular redox environment and tested the role of cytosolic ROS production for CaT alternans development. Finally, we put a focus on the role of ROS_m_ for the generation of CaT alternans by using mitochondria-directed antioxidants and experimentally probing elements of the mitochondrial antioxidant defense mechanisms, which could be potential pharmacological targets for alternans prevention and treatment.

## 2. Material and Methods

### 2.1. Ethical Approval

All aspects of animal husbandry, animal handling, anesthesia, surgery and euthanasia were fully approved by the Institutional Animal Care and Use Committee (IACUC protocol 22-027; approved 9 May 2022) of Rush University Chicago and comply with the National Institutes of Health’s Guide for the Care and Use of Laboratories Animals.

### 2.2. Chemicals, Solutions and Experimental Conditions

All chemicals were purchased from Sigma-Aldrich (St. Louis, MO, USA) unless stated otherwise. The external normal Tyrode (NT) solution was composed of (in mM) 135 NaCl, 5 KCl, 1 MgCl_2_, 2 CaCl_2_, 5 HEPES, 5 Na-HEPES and 10 D-glucose (pH 7.4, adjusted with 1 N HCl). Stock solutions of quercetin (50 mM), MnTBAP (25 mM), MitoTempo (40 mM), apocynin (7.5 mM), NOX2-inhibitor GSK2795039 (12 mM), NOX4-inhibitor GLX351322 (5 mM), S1QEL1.1 (7.5 mM), S3QEL2 (20 mM), auranofin (15 mM), 4-chloro-7-nitrobenzo-2-oxa-1,3 diazole (NBD-Cl; 10 mM) were prepared in DMSO. Stock solutions were diluted to final concentration in external NT solution. Corresponding amounts of DMSO were added to control solutions (final DMSO concentration (*v*/*v*) ≤ 0.2%). SS-31 (10 mM) was dissolved in deionized water. 4-Hydroxy-tempo (Tempol), dithiothreitol (DTT) and tert-butyl hydroperoxide (tBHP) solutions were prepared directly in NT. All experiments were performed at room temperature (20–22 °C).

### 2.3. Isolation of Atrial Myocytes

For the enzymatic isolation of left atrial myocytes, male New Zealand White rabbits (2–3 kg, 80 rabbits; Envigo, Indianapolis, IN, USA, and Charles River, Wilmington, MA, USA) were anesthetized with an intravenous injection of sodium pentobarbital (100 mg/kg) and heparin (1000 U/kg) 15 min before thoracotomy. The depth of the anesthesia was evaluated by foot pinch or checking corneal reflexes. Hearts were excised and washed in cold Ca-free NT solution supplemented with 1000 U/L heparin. Hearts were mounted on a Langendorff apparatus and retrogradely perfused via the aorta. After an initial 5–10 min perfusion with oxygenated Ca-free NT solution at 37 °C, the heart was perfused for ~20 min at 37 °C with oxygenated minimal essential medium Eagle MEM solution (Joklik’s modification, Sigma-Aldrich, product #M0518) containing 20 µM Ca and 22.5 µg/mL Liberase TH (Roche Diagnostic Corporation, Indianapolis, IN, USA). The MEM solution was supplemented with 2 mM sodium pyruvate, 10 mM taurine, 10 mM HEPES, 10 mM Na-HEPES, 23.8 mM NaHCO_3_, 50,000 U/L penicillin, 50 mg/L streptomycin and 40 U/L insulin; pH 7.4 (adjusted with 1N HCl). The left atrium was dissected from the heart and minced, filtered and washed in MEM solution with 50 µM Ca and 1% bovine serum albumin. Isolated cells were washed and kept in MEM solution with 50 µM Ca at room temperature (20–22 °C) and were used for experimentation within 8 h after isolation.

### 2.4. Cytosolic Ca ([Ca]_i_) Measurements

Cytosolic Ca transients (CaT) were recorded in intact left atrial myocytes with the fluorescent Ca indicators Cal-520/AM or Cal-590/AM (AAT Bioquest, Sunnyvale, CA, USA) using an IonOptix epifluorescence microscopy setup (IonOptix LLC, Westwood, MA, USA). Briefly, atrial myocytes were loaded with 10 μM Cal-520/AM or 5 µM Cal-590/AM in the presence of 0.05% Pluronic F-127 (Thermo Fisher Scientific, Waltham, MA, USA) in NT solution for 20 min. The loading solution was replaced by NT solution and cardiomyocytes were washed for 20 min to allow de-esterification of the Ca dye. Cells were plated on laminin (1 mg/mL) coated coverslips and continuously superfused with NT solution during the experiment. Myocytes were electrically field stimulated through a pair of platinum electrodes with electrical stimuli of voltage ~50% greater than the threshold for myocyte contraction [33]. Cal-520 fluorescence was excited at 488 nm by a Xe arc lamp and cytosolic CaTs were recorded at 517 nm. Cal-590 fluorescence was excited at 555 nm, and fluorescence emission was measured at 605 nm. Fluorescence emission signals (F) were background subtracted and normalized to diastolic [Ca]_i_ (F_0_), measured at the beginning of a recording. Changes of [Ca]_i_ are presented as F/F_0_.

### 2.5. ROS Measurements

Cellular ROS production was monitored with the ROS-sensitive fluorescent probe 5-(-6)-chloromethyl-2′,7′-dichlorohydrofluorescein diacetate (CM-H_2_DCFDA; Thermo Fisher Scientific, Waltham, MA, USA). Oxidation of CM-H_2_DCFDA by a variety of ROS including H_2_O_2_, peroxynitrite, nitric oxide, lipid hydroperoxides and O_2_^·−^ yields the fluorescent product dichlorofluorescein (DCF) [34]. For simultaneous measurements of ROS and CaT alternans, atrial myocytes were incubated with 10 μM CM-H_2_DCFDA and 5 μM of Cal-590/AM for 20 min and washed with NT solution for 20 min, followed by [Ca]_i_ and cellular ROS measurements. DCF fluorescence was excited at 488 nm, and fluorescence emission was recorded at 517 nm. Cellular ROS production in electrically stimulated cells was quantified as the rate of change of the background subtracted and normalized DCF signal (F/F_0_/s). F_0_ is the DCF fluorescence in unstimulated cells at the beginning of the recording.

### 2.6. CaT Alternans

CaT alternans was elicited by increasing the pacing frequency until stable CaT alternans was observed. Diastolic [Ca]_i_ before the large amplitude CaT was chosen as F_0_. The degree of CaT alternans was quantified as alternans ratio (AR), defined as AR = 1 − CaT_small_/CaT_large_, where CaT_small_ and CaT_large_ are the amplitudes of the small and large CaTs of a pair of alternating CaTs [35,36]. Accordingly, AR values vary on a continuum between 0 and 1. AR = 0 indicates no alternans, and AR = 1 indicates the highest possible degree of CaT alternans with every other stimulation failing to trigger a measurable CaT. CaTs were considered alternating when the beat-to-beat difference in CaT amplitude exceeded 10% (AR > 0.1) [37]. CaT amplitude (ΔF/F_0_) was measured as the difference in F/F_0_ measured immediately before the stimulation pulse and the peak of the CaT. To calculate average ARs, 7–20 consecutive CaT alternans pairs (14–40 individual CaTs) were analyzed.

### 2.7. Data Presentation and Statistics

Data were collected and analyzed using IonWizard 6.2 (IonOptix), OriginPro 2016 (OriginLab Corporation, Northampton, MA, USA) and GraphPad Prism 5 (GraphPad Software, San Diego, CA, USA). All summary data are presented as mean ± SEM for the indicated number (*n*) of cells or measurements (as specified) from *N* rabbits. For two-group comparisons, Student’s *t*-test for paired or Mann–Whitney U test for unpaired data was used. For multiple-group analysis ANOVA for repeated or single measurements with Tukey’s post-hoc test was used. Data were considered significant at *p* < 0.05.

## 3. Results

### 3.1. Correlation of Cellular ROS Production and CaT Alternans

Enhanced cellular ROS production has been linked to cardiac arrhythmias and contractile dysfunction in HF, myocardial infarction (MI), ischemia/reperfusion (IR)-injury, diabetes, hypertension and cardiac disorders associated with aging [38]. Furthermore, cardiac alternans is connected directly to cardiac arrhythmias and its prevalence is increased in HF [39]. We determined if cellular ROS production is related to CaT alternans development. For this purpose, atrial myocytes were loaded with the fluorescent indicators CM-H_2_DCFDA and Cal-590/AM to measure ROS production and CaT alternans simultaneously. Cells were stimulated by incrementally increasing pacing frequency between 0.5 and 2.5 Hz or until stable CaT alternans appeared (*N* = 23, *n* = 45 cells; Figure 1).

Figure 1A shows atrial myocyte CaTs and developing CaT alternans at increasing pacing frequencies simultaneously recorded with the rate of cellular ROS generation. Figure 1B shows the fraction of cells that develop CaT alternans at a given stimulation frequency and the degree of CaT alternans (alternans ratio) as a function of increasing stimulation rates. Across all stimulation frequencies the ROS production rate was ~26% higher in alternating cells compared to non-alternating cells and increased from 1.6 ± 0.2 to 2.2 ± 0.4 × 10^−3^ (F/F_0_/s) (Figure 1C). Figure 1D shows the ROS production rate and fraction of cells with CaT alternans as a function of stimulation frequency. At low stimulation frequency (0.5 Hz) cells failed to develop alternans. At 1 Hz CaT alternans appeared in a small fraction of cells (7 of 43 cells or 16%). At 1.5 Hz roughly half of the cells developed alternans (52% vs. 48% non-alternating cells). At 1 Hz and 1.5 Hz both groups showed comparable levels of ROS production, but ROS production was overall higher at 1.5 Hz compared to 1 Hz. Larger differences between the two groups were observed at higher pacing rates (2 Hz and 2.5 Hz). The fraction of alternating cells increased to 80–87%. The difference in ROS production between non-alternating and alternating cells became substantial. At 2 Hz ROS production rates increased from 0.9 ± 0.2 × 10^−3^ to 2.1 ± 0.6 × 10^−3^ (F/F_0_/s), and at 2.5 Hz, the difference became even more pronounced (0.8 ± 0.4 vs. 3.2 ± 1.5 × 10^−3^ F/F_0_/s). The small fraction of non-alternating cells at 2 and 2.5 Hz showed lower ROS production than cells at 0.5 Hz, supporting the notion that non-alternating cells tend to have a lower ROS production rate, and further emphasizing that increased ROS production correlated with a higher degree of CaT alternans. Overall, our results suggest that cellular ROS production contributes to CaT alternans development.

### 3.2. Oxidative Stress Facilitates CaT Alternans

To test further whether an oxidative environment affects pacing-induced CaT alternans, we exposed atrial cells to an exogenous source of ROS. We challenged cells with tert-butyl hydroperoxide, a stable form of H_2_O_2_ (tBHP, 1 mM) [40]. tBHP enhanced CaT alternans and increased AR by ~140% from 0.28 ± 0.04 in Ctrl to 0.67 ± 0.05 (*N* = 7, *n* = 19; *p* = 0.0001; Figure 2A). The opposite was observed when atrial cells were exposed to the thiol-reducing agent dithiothreitol (DTT, 1 mM) [41]. As shown in Figure 2B, DTT decreased AR by 39% from 0.52 ± 0.07 to 0.32 ± 0.07 (*N* = 2, *n* = 6; *p* = 0.0194). When DTT was applied in addition to tBHP, the enhancement of CaT alternans by tBHP was largely abolished (Figure 2C). The data demonstrate that oxidative conditions enhance the susceptibility of atrial cells to develop CaT alternans.

The involvement of ROS in the development of CaT alternans was further evaluated by testing the effects of the ROS scavenger quercetin [42] and the SOD mimetics MnTBAP [43] and 4-Hydroxy-tempo (Tempol) [44]. Quercetin (10 µM) [45] exposure of atrial myocytes suppressed pacing-induced CaT alternans and decreased CaT AR by 96% from 0.47 ± 0.05 to 0.02 ± 0.01 (*N* = 2, *n* = 6; *p* = 0.0007; Figure 3A). MnTBAP (50 µM) [43] lowered CaT AR by 46% (from 0.83 ± 0.11 to 0.45 ± 0.14; *N* = 2, *n* = 6; *p* = 0.0248; Figure 3B). An AR decrease was also obtained with Tempol (1.6 mM) [46] which significantly lowered CaT AR by 79% from 0.58 ± 0.09 to 0.12 ± 0.03 (*N* = 4, *n* = 8; *p* = 0.0029; Figure 3C). Overall, these data support the notion that an oxidative environment facilitates CaT alternans.

### 3.3. Role of NADPH Oxidases in Atrial CaT Alternans

The family of NADPH oxidases [47], of which NOX2 and NOX4 are the most abundantly expressed isoforms in cardiomyocytes [48,49], are known ROS producers in the cardiovascular system. We tested the NOX involvement in CaT alternans regulation. First, we applied the broadly acting NOX inhibitor apocynin (1 μM) [41] to cells showing stable CaT alternans. Apocynin failed to affect the degree of CaT alternans with an AR of 0.56 ± 0.09 in Ctrl and 0.57 ± 0.10 in the presence of apocynin (*N* = 3, *n* = 6; *p* = 0.8338; Figure 4A). Subsequently, we applied more specific inhibitors targeted to NOX2 and NOX4. NOX2 inhibitor GSK2795039 (5 μM) [50] was applied to cells showing stable CaT alternans (Figure 4B). Our results indicate that on average NOX2 inhibition did not affect the degree of CaT alternans, with AR = 0.40 ± 0.04 in Ctrl and 0.33 ± 0.09 in the presence of the inhibitor (*N* = 6, *n* = 13; *p =* 0.3596). The NOX4 isoform is expressed primarily in mitochondria in cardiac myocytes [51] and is generally assumed to be constitutively active [52,53]. Similar to NOX2 inhibition, NOX4 inhibition with GLX351322 (5 μM) [54] on average did not change AR (0.43 ± 0.05 in Ctrl vs. 0.48 ± 0.13, *N* = 5, *n* = 8; *p =* 0.6129; Figure 4C). The effects of NOX2 and NOX4 inhibitors are in line with the results obtained with apocynin, and the data suggest that NADPH-oxidases, including NOX2 and NOX4, do not play a key role in the development of pacing-induced CaT alternans.

### 3.4. Mitochondrial ROS (ROS_m_) and CaT Alternans

Under physiological conditions mitochondria are also an important source of ROS production through the single electron reduction of molecular oxygen during the process of oxidative metabolism (in addition to the aforementioned NOX4 activity), generating O_2_^·−^ which is subsequently dismuted to H_2_O_2_. Oxidative stress can occur when ROS production is increased or by depletion of the antioxidant defense [20]. Thus, to determine the role of ROS_m_ production as a trigger and modulator of the severity of CaT alternans, we tested the effect of the mitochondria-targeted antioxidant MitoTempo, a SOD mimetic of mitochondrial O_2_^·−^ [55,56]. MitoTempo (20 μM) [57] was tested at pacing frequencies between 0.5 and 2 Hz (Figure 5A,B). MitoTempo tended to decrease the degree of CaT AR at stimulation frequencies >1 (Figure 5A, bottom; *N* = 5, *n* = 9) compared to Ctrl (Figure 5A, top; *N* = 5, *n* = 10), and shifted the pacing frequency threshold for alternans to higher levels (Figure 5B). Since the MitoTempo effects did not reach statistical significance, we tested the effect of the mitochondrial cytochrome *c* (cyt *c*) peroxidase inhibitor SS-31 [58]. After stable CaT alternans was established, the application of SS-31 (20 μM) [59] decreased AR by 75% from 0.36 ± 0.05 in Ctrl to 0.09 ± 0.03 (*N* = 2, *n* = 6; *p* = 0.0003; Figure 5C), suggesting that SS-31 decreases ROS_m_ production by cyt c peroxidase and protects from alternans

Next, we investigated the role of ROS_m_ generated by the ETC. ETC complexes I (NADH: ubiquinone oxidoreductase) and III (ubiquinol: cyt *c* oxidoreductase) are known to be major sources of ROS from the ETC [10,60]. To investigate their role in CaT alternans modulation, we used pharmacological suppressors of O_2_^·−^-H_2_O_2_ production by complex I (S1QEL1.1, suppressor of site I_Q_ electron leak) [61] and by complex III (S3QEL2, suppressor of site III_QO_ electron leak) [62]. These compounds do not affect the electron flow or oxidative phosphorylation, unlike other commonly used inhibitors such as rotenone and antimycin A. Both S1QEL1.1 (Figure 5D) and S3QEL2 (Figure 5E) rescued CaT alternans. S1QUEL1.1 (5 μM) [61] decreased AR by ~36% from 0.47 ± 0.10 in Ctrl to 0.30 ± 0.10 (*N* = 3, *n* = 8; *p* = 0.0560). S3QEL2 (15 μM) [62] lowered AR by 57% from 0.45 ± 0.07 in Ctrl to 0.19 ± 0.08 (*N* = 8, *n* = 11; *p* = 0.0072). These findings indicate that the O_2_^·−^-H_2_O_2_ production by ETC complexes I and III has an important role in the development of CaT alternans. Together, these data indicate that ROS from various mitochondrial sources facilitates CaT alternans.

### 3.5. Impairment of Mitochondrial Antioxidant Defense Promotes CaT Alternans

Mitochondrial antioxidant defense mechanisms rely on the H_2_O_2_ scavenging capacity of the glutathione (GSH) and PRX/TRX-2 systems [15,16], and require the continuous availability of NADPH-reducing equivalents for GSH reductase and thioredoxin reductase-2 (TR2). To further investigate the role of ROS_m_ in the development of CaT alternans, we selectively inhibited the mitochondrial THD that provides the necessary NADPH pool for the reductase action of GPX and PRX [17,18]. AR was measured as a function of a stepwise increase in electrical stimulation frequency between 0.5 and 2 Hz (Figure 6A,B) under Ctrl (*N* = 5, *n* = 12) conditions and in the presence of the THD blocker NBD-Cl (4-chloro-7-nitrobenzo-2-oxa-1,3 diazole, 2.5 µM [63]; *N* = 5, *n* = 9). NBD-Cl enhanced the degree of CaT alternans over the entire range of pacing frequencies tested and lowered the pacing threshold for alternans (Figure 6B). Furthermore, inhibition of TR2 by auranofin (10 µM) [64,65] increased AR by 68% from 0.42 ± 0.08 in Ctrl to 0.71 ± 0.04 in the presence of auranofin (*N* = 3, *n* = 7; *p* = 0.0004; Figure 6C). These results are consistent with the notion that depletion of the antioxidant pool facilitates CaT alternans.

## 4. Discussion

In the present study we tested the hypothesis that ROS_m_ plays a critical role in the development of atrial calcium alternans. The main findings in support of this hypothesis are as follows:(1)the cellular redox environment modulates pacing-induced CaT alternans development: exogenous ROS (applied in the form of tBHP) enhanced CaT alternans whereas ROS scavengers and reducing agents decreased AR;(2)cellular ROS production is increased during CaT alternans and the degree of alternans and rate of ROS production correlate with the pacing rate;(3)pharmacological inhibitors of NADPH oxidases, including specific inhibitors of NOX2 and NOX4, failed to affect alternans and excluded an important role of these specific ROS sources for alternans regulation;(4)several lines of evidence point to the important role of ROS_m_ for the modulation of alternans: MitoTempo, a mitochondrial SOD mimetic, shifted the alternans pacing threshold to higher frequencies and decreased the degree of alternans; in the presence of the inhibitor of cyt *c* peroxidase SS-31 and during the inhibition of ROS production at ETC Complexes I and III with the small molecules S1QEL1.1 and S3QEL2, AR decreased or alternans even disappeared; the impairment of mitochondrial antioxidant defense by THD inhibition with NBD-Cl and the inhibition of TR2 with auranofin enhanced alternans.

### 4.1. Correlation of Pacing-Induced ROS Production and CaT Alternans Development

It is well known that ROS production in cardiac myocytes is induced by increasing the beating rate [66] and that cardiac alternans can be induced by pacing. Furthermore, the degree of alternans increases with increasing beating frequency [67]. Here we show for the first time that pacing-induced CaT alternans in atrial myocytes is tightly linked to ROS production, and the degree of alternans and rate of ROS production both correlate with pacing frequency, although we noticed a fair degree of cell-to-cell variability of the rate of cellular ROS generation (cf. Figure 1A,D). Across all stimulation frequencies tested between 0.5 and 2.5 Hz, the rate of ROS production was 26% higher in cells showing CaT alternans compared to non-alternating cells (Figure 1C). The fraction of cells revealing CaT alternans steadily increased with increasing pacing rates together with the rate of ROS generation (Figure 1B), whereas at high pacing rates (2–2.5 Hz) the small fraction of remaining cells that failed to develop CaT alternans showed the lowest level of ROS production.

### 4.2. ROS_m_ Production and CaT Alternans in Atrial Myocytes Are Interconnected

We previously showed that altered mitochondrial function induces CaT alternans. Alternans can be facilitated either by the inhibition of the mitochondrial energy metabolism (through the inhibition of Ca-dependent dehydrogenases of the Krebs cycle, complexes of the ETC, and the collapse of the mitochondrial membrane potential, ΔΨ_m_) or through interference with mitochondrial Ca cycling [67,68]. These disturbances impair the Krebs cycle turnover rate which in turn modulates ROS_m_ production and elimination [69], which can undermine mitochondrial antioxidant defense and lead to oxidative stress.

Here we showed that CaT alternans coincides with higher rates of cellular ROS production (Figure 1). We used the fluorescent probe CM-H_2_DCFDA to measure cellular ROS. CM-H_2_DCFDA is a suitable tool to screen for any cellular ROS generation because it reacts with a number of different ROS that can originate from various cellular sources. While suitable as a screening tool, the probe cannot identify specific ROS or a specific ROS source, for example ROS_m_. To investigate the role of ROS_m_ in the generation of CaT alternans, we employed a strategy where we probed pharmacologically several targets of ROS_m_ generation directly. One of the strategies entailed the use of the mitochondria-targeted SOD mimetic MitoTempo. MitoTempo decreased the degree of CaT alternans, suggesting that mitochondrial O_2_^·−^ production is involved. The alleviating effects of S1QEL1.1 and S3QEL2 on CaT alternans further confirmed the involvement of ROS_m_ production. S1QEL1.1 is a small molecule and selective suppressor of the O_2_^·−^/H_2_O_2_ production of site I_Q_, which is active during the reverse electron transport of Complex I. S1QEL1.1 has been used to show that site I_Q_ contributes actively to O_2_^·−^/H_2_O_2_ production in a cardiomyocyte cell model of ROS-induced endoplasmic reticulum stress and to protect against IR-injury in the perfused heart. In this model, S1QEL1.1 improved cardiac function and reduced infarct size [61]. Remarkably, S1QEL1.1 does not interfere with forward electron transport, unlike rotenone, thus avoiding the NAD^+^ pool becoming highly reduced which leads to oxidative stress [15,70]. Unlike site I_Q_ which releases O_2_^·−^ into the mitochondrial matrix, site III_QO_ releases about half of the O_2_^·−^ produced into the intermembrane space and the other half into the matrix. Subsequently, after its dismutation, a significant amount of H_2_O_2_ and O_2_^·−^ is released into the cytosol [71]. Due to the proximity of mitochondria and RyR2 and SERCA, ROS_m_ production by site III_QO_ can play an important role in the modulation of Ca cycling, and therefore for the susceptibility of atrial myocytes to develop alternans. RyR2s and SERCA are sensitive to oxidation by H_2_O_2_ [72]. Oxidation of RyR2 thiol groups from cysteine residues increases the open probability of the RyR2s and promotes SR Ca leak. On the other hand, SERCA activity can be inhibited by ROS, decreasing the Ca uptake rate [22,73]. Thus, these two mechanisms lead to an increase in diastolic [Ca]_i_ that favors CaT alternans development [3]. In support of this notion, we observed significant protection against CaT alternans by S3QEL2 treatment, suggesting the involvement of the ETC site III_QO_.

To further determine how ROS-mediated mitochondrial dysfunction underlies CaT alternans, we tested the effects of SS-31, a Szeto-Schiller peptide that has been shown to improve mitochondrial function impaired by oxidative stress [74,75]. Mitochondrial O_2_^·−^ overproduction plays a key role in the development of heart diseases such as MI, hypertensive cardiomyopathy, IR-injury and HF where SS-31 treatment has improved mitochondrial function, cardiac remodeling and left ventricular function, and has decreased mortality [12,15,38,76,77]. SS-31 binds by electrostatic and hydrophobic interactions selectively to cardiolipin (CL), a phospholipid that exclusively resides in the internal mitochondrial membrane. CL is a platform for electrostatic interaction with cyt *c* to facilitate the electron transfer between Complex III and IV, and CL plays a key role in stabilizing cristae architecture and the stabilization of respiratory complexes into supercomplexes [74,78]. Although the mechanisms are not completely understood, in the presence of H_2_O_2_ cyt *c* can undergo a CL-mediated conformational change that promotes its conversion from an electron carrier into a peroxidase/oxygenase. As a consequence, subsequent CL peroxidation disrupts mitochondrial supercomplexes and impairs mitochondrial bioenergetics and ATP synthesis, leading to mitochondrial dysfunction [58,79,80]. SS-31 modulates the interaction between CL and cyt *c* and inhibits cyt *c* peroxidase activity, thereby decreasing H_2_O_2_ production, preventing mitochondrial permeability transition pore (PTP_m_) opening, and enhancing the electron transfer and ATP synthesis [58,81]. We previously showed that PTP_m_ plays a role in the development of CaT alternans [67,68]. Thus, it is conceivable that the rescue of CaT alternans by SS-31 observed here (Figure 5C) may result not only from the improvement of mitochondrial function but also from the inhibition of ROS-induced PTP_m_ opening [81,82]. Other mechanisms by which SS-31 may contribute to the decrease in ROS_m_ are by rescuing the decreased activity of Complex III arising from CL peroxidation [83] and by improving electron transfer which increases ATP availability. It has been shown that compromised mitochondrial ATP production interferes with Ca handling by different mechanisms, including diminished SERCA activity due to low ATP availability, decreased allosteric activation of RyR2s by ATP, and NCX-mediated Ca overload, all factors that tend to facilitate CaT alternans [22,84,85]. Furthermore, restoring mitochondrial function may improve mitochondrial Ca uptake and sequestration, which protects from CaT alternans as well [67].

We also explored the role of the mitochondrial GSH/GPX and PRX/TRX-2 systems which represent two key elements of cellular antioxidant protection. Inhibition of TR2 by auranofin eliminates an important element of the PRX/TRX-2-H_2_O_2_ scavenging system [64] and thus of the mitochondrial antioxidant defense. In the presence of auranofin, atrial myocytes developed strong CaT alternans with high ARs of ~0.7 (Figure 6C). The inhibition of TR2 activity by auranofin alters the mitochondrial redox balance and increases H_2_O_2_ extrusion in isolated energized liver mitochondria [65] as well as in cardiac mitochondria and cardiomyocytes [64]. In cardiomyocytes, the increase in mitochondrial H_2_O_2_ translated into an increase in H_2_O_2_ in the cytosol without altering NADPH or GSH levels [64]. In addition, TR2-deficient cardiomyoblasts display a sharp increase in mitochondrial H_2_O_2_ in response to a prooxidant challenge, highlighting its role in maintaining the redox balance [32].

A previous microarray study showed that GPX1, which is localized in both the cytosol and mitochondria [86], is downregulated in atrial appendages from AF patients [87]. To investigate the role of the GSH/GPX pathway for CaT alternans regulation, we employed butyl malonate to block mitochondrial GSH uptake [63]. Since GSH is synthesized in the cytosol [88], under these conditions GSH levels increase in the cytosol and increase the reducing power. However, we did not observe any significant effect of butyl malonate on the AR of pacing-induced CaT alternans, suggesting that the GSH antioxidant system, contrary to the PRX/TRX-2 system, plays a lesser role in CaT alternans control.

Mitochondrial antioxidant defense mechanisms depend on the availability of NADPH, which is mainly supplied by THD. THD regenerates NADPH from NADH to restore oxidized GSH and TRX-2 [17,18,32]. We demonstrate that inhibition of THD activity by NBD-Cl in atrial myocytes promotes CaT alternans (Figure 6A,B), thus confirming the role of the mitochondrial antioxidative capacity on CaT alternans regulation. Dey et al. showed that silencing of THD in cardiomyoblasts severely compromised their H_2_O_2_ scavenging capacity [32]. Expression and activity of THD and TR2 appear to play an important role for the susceptibility of CaT alternans and the development of AF in vivo; however, future studies will have to determine the underlying mechanisms.

## 5. Conclusions

Given the important role of ROS_m_ in the pathogenesis of cardiovascular diseases, it has been proposed to target antioxidants or antioxidant enzymes to mitochondria as preventive and therapeutic strategies [15,17,38,57,76,77]. However, cytosolic sources of ROS have also been implicated [27,89,90]. Serving in a protective role, mitochondria have been proposed to act as ROS sinks to eliminate not only ROS_m_, but also cytosolic or extracellular H_2_O_2_ [18,32]. We conclude that ROS_m_ production by ETC Complexes I and III, and thus the mitochondrial redox environment, are key determinants for the pathogenesis of the pro-arrhythmic condition of CaT alternans. Therefore, modulation of the mitochondrial redox state and mitochondrial antioxidative defense mechanisms represents a potential avenue for targeted therapies for alternans-associated cardiac arrhythmias, including AF.

## Figures and Tables

**Figure 1 biomolecules-14-00144-f001:**
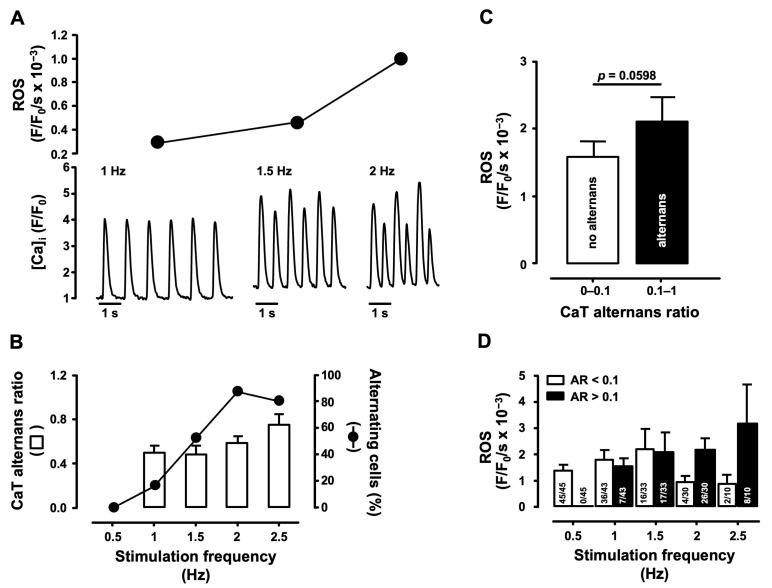
Cellular ROS production during CaT alternans. (**A**) Simultaneous [Ca]_i_ and cellular ROS generation rate measurements in an atrial myocyte during increasing pacing frequencies (*N*/*n* = 23/45). (**B**) CaT AR and fraction of alternating cells as a function of stimulation frequency (0.5–2.5 Hz). (**C**) Average ROS production rate in non-alternating (AR = 0–0.1; 103 measurements) and alternating cells (AR = 0.1–1; 58 measurements). Statistical analysis was performed using the Mann–Whitney U test. (**D**) ROS production rate in non-alternating and alternating cells as a function of stimulation frequency.

**Figure 2 biomolecules-14-00144-f002:**
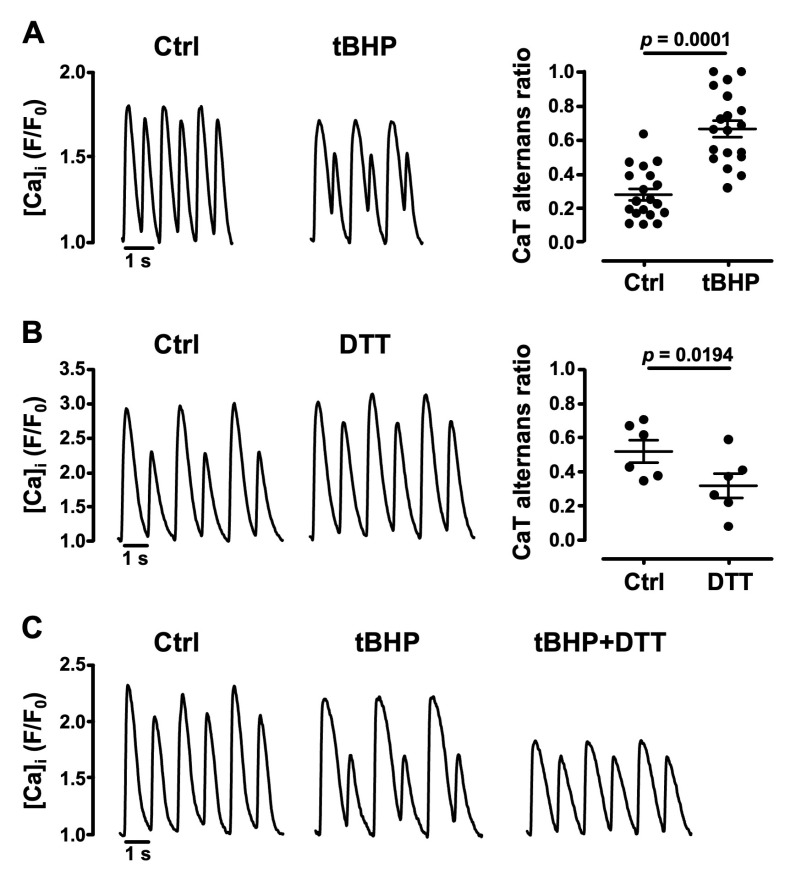
CaT alternans modulation by redox environment. (**A**) CaT alternans recordings (left) from atrial cells before (Ctrl) and during exposure to tert-butyl hydroperoxide (tBHP, 1 mM), paced at 1.7 Hz. Right: mean ± SEM and individual AR measurements in Ctrl and H_2_O_2_ (*N*/*n* = 7/19). (**B**) CaT alternans in Ctrl and during dithiothreitol (DTT, 1 mM) exposure (left), paced at 1 Hz. Right: mean ± SEM and individual AR measurements in Ctrl and DTT (*N*/*n* = 2/6). (**C**) CaT alternans in Ctrl, in the presence of tBHP and tBHP + DTT recorded from the same atrial myocyte, paced at 1 Hz. Statistical analysis (**A**,**B**) performed using Student’s *t*-test for paired data.

**Figure 3 biomolecules-14-00144-f003:**
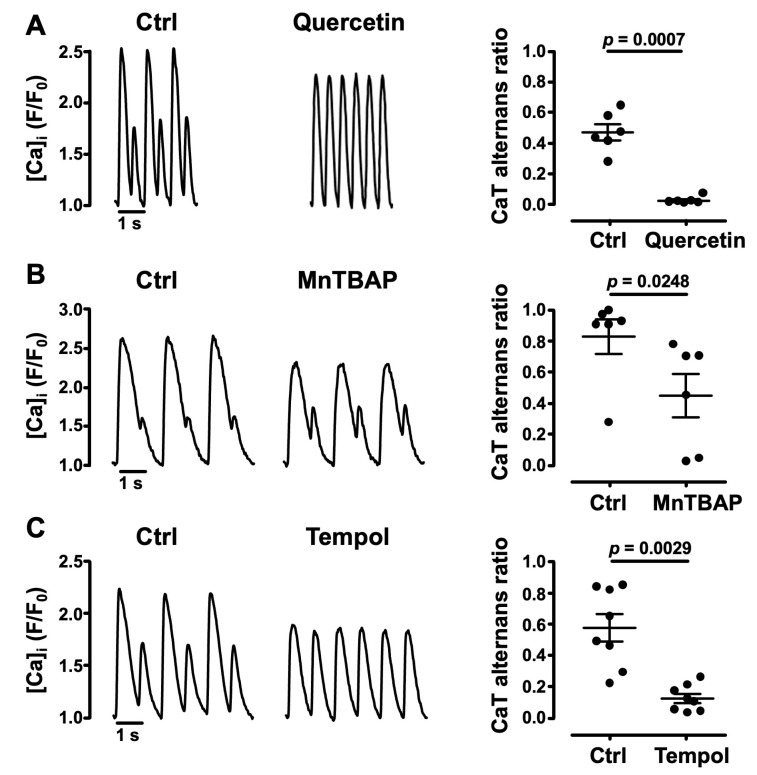
Effect of antioxidants on CaT alternans. CaT alternans recordings from atrial myocytes (left panels) before (Ctrl) and during the application of (**A**) superoxide scavenger quercetin (10 µM), paced at 2.1 Hz; (**B**) SOD mimetic MnTBAP (50 µM), paced at 1.1 Hz; and (**C**) SOD mimetic Tempol (1.6 mM), paced at 1.1 Hz. Right panels: mean ± SEM and individual AR measurements in Ctrl and quercetin (*N*/*n* = 2/6), MnTBAP (*N*/*n* = 2/6), and Tempol (*N*/*n* = 4/8), respectively. Statistical analysis performed using Student’s *t*-test for paired data.

**Figure 4 biomolecules-14-00144-f004:**
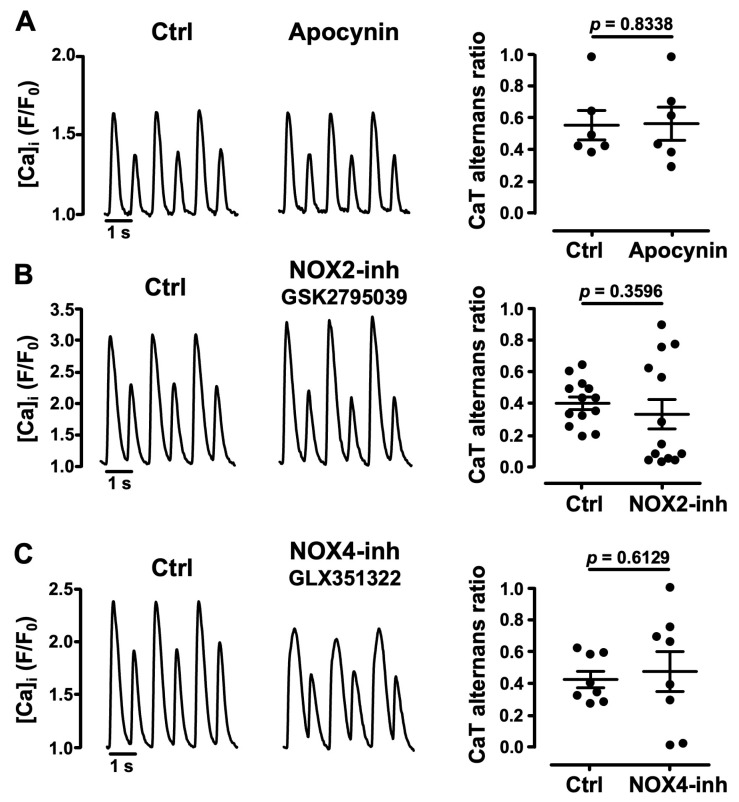
Effects of inhibition of NADPH oxidases on CaT alternans. (**A**) CaT recordings from atrial cells (left panels) before (Ctrl) and during the application of (**A**) Apocynin (1 µM), paced at 1.1 Hz; (**B**) NOX2 inhibitor GSK2795039 (5 µM), paced at 1.25 Hz; and (**C**) NOX4 inhibitor GLX351322 (5 µM), paced at 1.3 Hz. Right panels: mean ± SEM and individual AR measurements in Ctrl and apocynin (*N*/*n* = 3/6), GSK2795039 (*N*/*n* = 6/13) and GLX351322 (*N*/*n* = 5/8). Statistical analysis performed using Student’s *t*-test for paired data.

**Figure 5 biomolecules-14-00144-f005:**
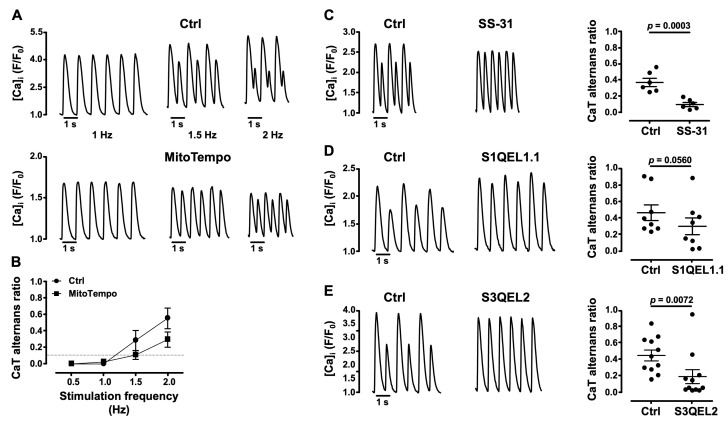
Effect of inhibition of ROS_m_ production on CaT alternans. (**A**) CaT recordings from atrial myocytes during electrical pacing at 1, 1.5 and 2 Hz in Ctrl and in the presence of the mitochondrial SOD mimetic MitoTempo (20 µM). Cells were incubated with MitoTempo for 5 min. (**B**) Average ARs in Ctrl (*N*/*n* = 5/10) and MitoTempo (*N*/*n* = 5/9) as a function of stimulation frequency (0.5–2 Hz). The horizontal dashed line marks the alternans threshold (AR > 0.1). (**C**–**E**). CaT alternans recordings (left panels) before (Ctrl) and during the application of (**C**) cyt *c* peroxidase inhibitor SS-31 (20 µM), paced at 2 Hz; (**D**) S1QEL1.1 (suppressor of site I_Q_ electron leak from mitochondrial Complex I; 5 µM), paced at 1.1 Hz; and (**E**) S3QEL2 (suppressor of site III_QO_ electron leak of mitochondrial Complex III; 15 μM), paced at 1.3 Hz. Right panels (**C**–**E**): mean ± SEM and individual AR measurements in Ctrl and SS-31 (*N*/*n* = 2/6), S1QEL1.1 (*N*/*n* = 3/8) and S3QEL2 (*N*/*n* = 8/11). Statistical analysis performed using Student’s *t*-test for paired data.

**Figure 6 biomolecules-14-00144-f006:**
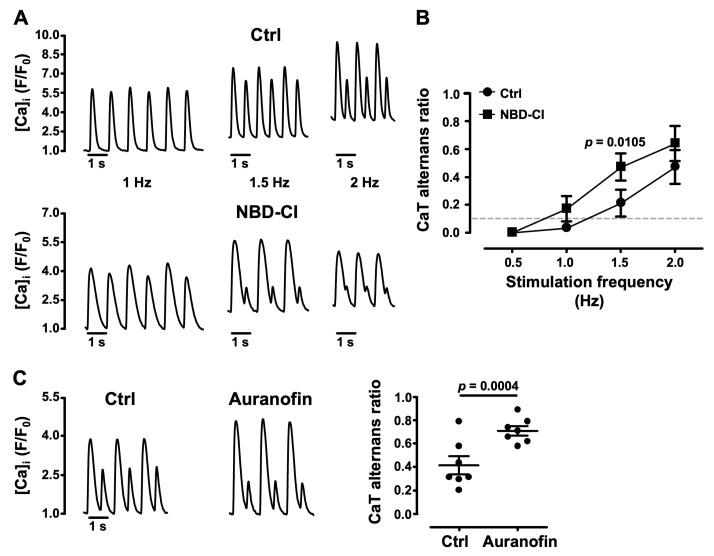
Effect of impairment of mitochondrial antioxidant defense on CaT alternans. (**A**) CaT recordings from atrial cells during electrical pacing at 1, 1.5 and 2 Hz in Ctrl (*N*/*n* = 5/12) and in the presence of the mitochondrial transhydrogenase (THD) inhibitor NBD-Cl (4-chloro-7-nitrobenzo-2-oxa-1,3 diazole; 2.5 µM; *N*/*n* = 5/9). (**B**) Average ARs in Ctrl and NBD-Cl as a function of stimulation frequency (0.5–2 Hz). The horizontal dashed line marks the alternans threshold (AR > 0.1). Statistical analysis performed using ANOVA. (**C**) Left: CaT alternans in Ctrl and during application of the mitochondrial TR2 inhibitor auranofin (10 μM), paced at 1.5 Hz. Right: mean ± SEM and individual AR measurements in Ctrl and auranofin (*N*/*n* = 3/7). Statistical analysis performed using Student’s *t*-test for paired data.

## Data Availability

The data presented in this study are available on request from the corresponding author.

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
