# Peer review of "Role of Mitochondrial ROS for Calcium Alternans in Atrial Myocytes"

_biomolecules, 2024, doi:10.3390/biom14020144_

Round 1

Reviewer 1 Report

Comments and Suggestions for Authors

This is an interesting study dissecting the dependance of Ca transients alternans on mitochondrial ROS. The analyses are carefully conducted and globally conclusions are supported by the data.

Comments.

Although most of the data is clear, there are two Figures (Fig. 5B and 6B), where statistics are not shown and it is not clear whether the claim is real, although the trend is clear. For example, the authors write that Mitotempo decreased the degree of Ca transients alternans at stimulation frequencies > 1Hz, but the figure does not show clear effect at 1.5 Hz, and the statistical significance at 2 Hz is not given. Moreover, inhibiting mitochondrial ROS production with mitotempo do not completely abolish alternans, thus other mechanisms are producing them. Please discuss.

Another concern is the mechanism of alternans. ATP production and its utilization by SERCA is briefly mentioned in Discussion, but a bit more explanations would be welcomed, as well as other targets of ROS that can participate in alternans.

Minor: The term “improving” when referring to alternans is confusing as one may think that favors alternans, or the opposite, as alternans are proarrhythmogenic.

Reviewer 2 Report

Comments and Suggestions for Authors

Authors performed experiments on isolated atrial myocytes to test the correlation between ROS production and CaT alternans under pacing. Authors further deepen the role of mitochondrial ROS and the mitochondrial antioxidant defences in CaT alternans susceptibility. 

I would like to point out to the Authors that the relation between ROS and CaT alternans in atrial fibrillation has been already proposed and investigated in literature. In order to improve the novelty of the manuscript, I suggest to the Authors to focus mainly on 1) the role of mitochondrial ROS, 2) the different response between cytosolic and mitochondrial ROS inhibitors and 3) the role of the mitochondrial antioxidant system.  

In general, the study has been properly conducted, however some points need to be clarified and other experiments should be performed to improve the novelty of the study as suggested above.

Major points:

- Since other works support the idea of a relation between ROS and CaT alternans in AF, references can be improved. 

- CM-H2DCFDA is a general probe for the whole cellular ROS. In order to investigate the specific contribution of mitochondrial ROS to the CaT alternans ratio, Mitosox can be used as a specific probe for mitochondrial superoxide.

- In the materials and methods section, Authors stated that CM-H2DCFDA fluorescence was measured after 20 minutes of incubation. Is this time respected also in the experiments in which Authors measured ROS and CaT alternans simultaneously?

- In figure 1D, ROS generation analysis was conducted from 0.5 to 2.5 Hz, while in figure 1A the 2.5 Hz is missing. Moreover, why the scale on the y axis is different between the two figures?

- Authors showed, in both figure 1A and 1B, a big difference in the graph slope between 0-0.4 and 0.4-1 for both the CaT alternans ratio and the ROS generation. To split the "alternating cells" group in these two ranges, instead of a single range 0.1-1, would allow, maybe, to obtain a statistical significance.

- Data showed in figure 1D at 1.5 and 2 Hz do not support the sentences at lines 190-193 and 335-336. Why at 1.5 and 2 Hz the ROS for alternating cells are the same? Why, instead, the ROS for not alternating cells are stable until 1.5 Hz and decrease from 2 Hz?

- The CaT alternans "controls" seem to be different among the experiments with inhibitors and ROS scavangers. Authors should indicate if myocytes were subjected to pacing and the related Hz for each experiment.      

- In the different experiments with inhibitors and ROS scavangers, how did Authors decide the concentration of the compound to test? Authors performed tritation experiments or concentrations are referred to a paper? In case, please add the related citation.  

- Authors propose to sustain the mitochondrial antioxidative defenses as a potential targeted therapy. If Authors were meaning the innate antioxidant enzymes expressed in our cells, I want to point out that Authors proved that the inhibition of these enzymes worsen the CaT alternans ratio but they did not prove a positive effect of an increased level or activity of these antioxidant enzymes. Are there any evidence in literature of a reduced expression or activity of the mitochondrial antioxidant enzymes in AF? Experiments aimed in increasing the level or the activity of these enzymes to prevent CaT alternans would support the conclusion.  

Minor points:

- line 59: a not formatted citation

- Specify to what "N" and "n" refer to

- line 182: is 7/43 not 36/43

- in figure 1: statistical significance is not reported

- in figure 1C the color legend is missing 

- in figure 2C, the related graph is missing

Round 2

Reviewer 2 Report

Comments and Suggestions for Authors

The authors addressed nearly all of the suggested concerns by adding the requested informations and revising certain concepts.

Regarding the use of CM-H2DCFDA as a probe for measuring cellular ROS, limitations concerning its selectivity, quantification, linearity, and susceptibility to artifacts remain. The authors themselves reported that they "encountered a fair amount of cell-to-cell variability in the ROS production rate measurements, especially at low pacing frequencies". Alternative probes such as Boronate probes, Amplex red, or Mitosox (with appropriate controls) may serve as more specific probes for specifically detecting mitochondrial ROS, a crucial aspect for a paper focused on the role of mitochondrial ROS.

In relation to the points discussing the different scales in the ROS measurement graphs and the similar ROS levels in various experiments in Figure 1, this variability could be, in some points, confounding.

Regarding Figure 2, graphs for A and B are present on the right side of the figure, but the graph for C is missing.
